# The Role of HSP47 in Thrombotic Disorders: Molecular Mechanism and Therapeutic Potential

**DOI:** 10.3390/cimb47040283

**Published:** 2025-04-17

**Authors:** Minodora Teodoru, Oana-Maria Stoia, Maria-Gabriela Vladoiu, Alexandra-Kristine Tonch-Cerbu

**Affiliations:** 1Medical Clinical Department, Faculty of Medicine, “Lucian Blaga” University, 550024 Sibiu, Romania; minodora.teodoru@ulbsibiu.ro (M.T.); oana.stoia@ulbsibiu.ro (O.-M.S.); 2County Clinical Emergency Hospital of Sibiu, 550245 Sibiu, Romania

**Keywords:** HSP47, thrombosis, platelet aggregation, collagen stability, therapy

## Abstract

This review aims to analyze the role of heat shock protein 47 (HSP47) in thrombosis and evaluate its potential as a therapeutic target for deep vein thrombosis (DVT). A systematic literature review was conducted using PubMed, Scopus, and Web of Science to identify studies on HSP47, thrombosis, and collagen, selecting only relevant and methodologically rigorous articles. HSP47 regulates platelet function and collagen interaction, playing a key role in deep vein thrombosis (DVT). HSP47, known for stabilizing collagen, also improves platelet–collagen binding and thrombus formation. In addition, reduced HSP47 levels reduce platelet adhesion, resulting in reduced thrombus formation, while inhibitors that target HSP47 decrease platelet aggregation in animal models. Naturally low levels of HSP47 during prolonged immobility are also found in hibernating mammals, such as bears, and are associated with reduced formation of thrombi, indicating a possible natural mechanism of thrombo-protection. This observation could inform new therapeutic approaches. Current studies use in vitro platelet aggregation assays, flow chamber assays, and collagen binding studies to investigate the role of HSP47 in clotting. This review aims to synthesize existing evidence to better understand HSP47’s role in clot formation and explore its potential as a target for novel DVT therapies.

## 1. Introduction

Heat shock proteins (HSPs) are crucial molecular chaperones that facilitate proper protein folding and prevent aggregation under stress conditions, thereby maintaining cellular proteostasis. Among these, heat shock protein 47 (HSP47) is specifically involved in the folding and secretion of collagen, binding to procollagen to ensure its stability during the maturation process in the endoplasmic reticulum (ER). While its fundamental role in collagen biology is well-documented, emerging evidence suggests that HSP47 may also significantly influence thrombosis, particularly in the context of deep vein thrombosis [1,2,3,4].

HSP47, or SERPINH1, is a collagen-specific molecular chaperone of the serpin superfamily, which uniquely does not possess protease inhibitory function. The serpin fold is further characterized with three β-sheets bestowing stability, nine α-helices providing flexibility, and a modified reactive center loop with a strong penchant for binding collagen (RDEL sequence at the sulfonated C-terminus), which help keep it within the endoplasmic reticulum. HSP47 is synthesized in the endoplasmic reticulum of collagen-producing cells, including fibroblasts and myofibroblasts, and is encoded by the SERPINH1 gene located on chromosome 11q13.5 to transcribe in the nucleus and translate on ER-bound ribosomes. It stabilizes procollagen’s triple-helix structure and is expressed in a heat-shock and collagen demand-dependent manner, rendering it important in normal physiology and when diseases develop, such as fibrosis [5].

Although it has beneficial roles, opposing evidence exists in the context of HSP47 in disease pathology. HSP47 dysregulation has been shown in some studies to aggravate the inflammatory response, thereby resulting in the progression of disease rather than resolution thereof. Similarly, high activity levels of HSP47 can result in the aggregation of misfolded proteins, causing ER stress and promoting inflammation pathways. Such inflammatory response can establish a pro-thrombotic state, which aggravates the risk of diseases such as DVT [1,5].

HSP47, thus, has distinct roles, mediating collagen stability but also, when dysregulated, inflammatory potential. Understanding these dynamics will be important for evaluating HSP47 as a therapeutic target, as interventions must navigate the entangled history of these multifunctional proteins and will therefore need to augment its protective roles while averting any pernicious effects. Therefore, further studies are necessary to delineate the numerous functions of HSP47 in maintaining cellular homeostasis and its potential roles in pathologies [6,7,8,9].

This review aims to explore the molecular mechanisms underlying the role of HSP47 not only in collagen dynamics but also in thrombus formation and stability. It will highlight key findings and studies that link HSP47 to coagulation processes and potential therapeutic targets for preventing thrombosis. By examining both aspects, this review seeks to provide insights into the dual roles of HSP47 in connective tissue disorders and its implications for vascular health.

To provide a concise overview of the key themes and findings discussed in this text, the following table summarizes the critical roles, therapeutic potential, and broader implications of HSP47 across various biological contexts and disease mechanisms (Table 1).

## 2. Background

A systematic literature search was performed in PubMed, Scopus, and Web of Science to retrieve studies related to HSP47, thrombosis, and collagen. The search terms were “HSP47”, “deep vein thrombosis”, “platelet aggregation”, “collagen stability”, “thrombosis therapy”. Articles were selected based on their topical relevance, recency, and methodological rigor.

Included studies provided experimental or clinical data on the role of HSP47 in thrombosis. Exclusion criteria were applied to remove studies that had no direct evidence of HSP47 involvement or had incomplete methodologies.

Data were extracted from eligible studies with regard to experimental models, effects on platelet function, effects on thrombus formation, and/protocols on HSP47 transduction. Physio-pathological effects were compared by analyzing animal and human studies.

Relevant in vitro and in vivo studies examining platelet aggregation, collagen–HSP47 interactions, and thrombus formation were summarized. Mouse models and hibernating animal studies were examined to understand natural thrombus resistance mechanisms.

## 3. Main Text

### 3.1. Introduction to Heat Shock Proteins

Heat shock proteins (HSPs) play an essential role as molecular chaperones in the cellular stress response, helping to maintain correct protein folding and preventing aggregation under stress conditions like heat, inflammation, or hypoxia. HSP47, a collagen-specific molecular chaperone, is particularly associated with collagen stability and vascular integrity. Abnormal HSP47 expression leads to pathological collagen synthesis and extracellular matrix (ECM) disruption associated with DVT pathogenesis [6,10].

HSP47 is an endoplasmic reticulum (ER)-resident molecular chaperone for collagens with the vital role of stabilizing procollagen during synthesis through binding of polypeptide chains to avoid misfolding and ensure proper triple-helix formation. It does this in all cells that synthesize collagen such as fibroblasts and endothelial cells but is tissue-specific for various types of collagens. In blood vessels, HSP47 helps vascular fibroblasts to synthesize types I and III of collagen—found in the adventitia and media of vessels, respectively—providing them with strength and elasticity as well as helping endothelial cells and pericytes to synthesize type IV of collagen for the basement membrane. Types I and III of collagen, expressed in vessel subendothelial injury, cause aggregation of platelets for hemostasis with HSP47’s role in their synthesis indirectly contributing to the process [4,11,12]. Procollagen is then channeled to the Golgi where HSP47 is released and recycled back to the ER through its RDEL motif for mature collagen formation vital for vascular health and blood clotting [13,14,15].

#### 3.1.1. Ensuring Protein Integrity: The Functions of HSP47 and Chaperones in Endoplasmic Reticulum Stress

HSP47 is a specialized chaperon for procollagen and is indispensable for correct folding and assembly. Such specificity reduces endoplasmic reticulum (ER) stress due to the accumulation of misfolded proteins. On the other hand, an imbalance of procollagen synthesis overwhelms HSP47, which causes excessive ER stress and inflammation, which lead to diseases like cardiovascular disease and pulmonary fibrosis [1,4,11].

Previous research has shown that HSP47 is capable of binding newly synthesized procollagen chains, thus helping align them properly for triple-helix formation in the ER. In the event of misfolding, HSP47 pools misfolded proteins in the ER, activating quality control mechanisms that are frequently composing obstinate ER stress and inflammation. Most studies of HSP47’s role in the heart are conducted in animal models, which if small in scope, may not fully account for potential compensatory pathways through the HSP47 pathways [1,12,13].

#### 3.1.2. The Folding Process of Procollagen

Procollagen exemplifies how it engages with ER enzymes and chaperones during folding. It consists of trimeric regions separated by a triple-helical forming region. If folding is hindered, the C-propeptides associate with BiP, leading to degradation. Folded C-propeptides then form trimers or associate with PDI. After trimerization, the triple-helical domain folds, depending on proline hydroxylation. Trimeric non-triple-helical molecules interact with P4H. Finally, the protein is transported to the Golgi apparatus, forming higher-order aggregates. HSP47 is a crucial mediator in the procollagen folding process within the endoplasmic reticulum (ER), necessary for the formation and stability of the collagen triple helix, which is vital for tissue mechanical strength and function. HSP47 specifically binds to the C-propeptides of nascent procollagen chains during the late synthesis phase in the ER, promoting the optimal alignment of proα1 and proα2 chains to ensure stable triple helical formation. This chaperone function prevents premature aggregation of procollagen molecules [5,16,17].

Once procollagen is properly folded, HSP47 facilitates its transport from the ER to the Golgi apparatus. In cases of improper folding, HSP47 retains misfolded procollagen in the ER, initiating quality control pathways to manage these proteins. Deregulation of HSP47 activity due to genetic mutations, overexpression under stress, or inadequate cellular responses can lead to misfolded procollagen. This misfolding may overwhelm the ER’s capacity, resulting in chronic ER stress [1,14].

HSP47, the ER chaperone, stabilizes the triple-helix structure of procollagen and prevents misfolding during synthesis but does not directly facilitate transport to the Golgi. Following proper folding, procollagen is transported to the Golgi via vesicle-mediated transport (COPII vesicles), where HSP47 is released typically due to pH change and recycles back to the ER via its RDEL motif, while procollagen is further processed. This is repeated in all tissues that synthesize collagen, including cells like fibroblasts, myofibroblasts, endothelial cells, osteoblasts, and chondrocytes, although different collagens are synthesized, for example, types I and III from fibroblasts and type IV from endothelial cells in blood vessels. Though universal, the relevance and types of collagen vary with each tissue’s function, such as vascular hemostasis or skin structure [13,14,18].

The consequences of elevated ER stress include activation of the unfolded protein response (UPR), which can trigger inflammation, cell apoptosis, or fibrosis—factors critical in various disease processes [19,20].

Abnormalities in collagen synthesis and assembly caused by HSP47 dysregulation can lead to alterations in the vascular extracellular matrix (ECM) and blood vessel dysfunction. Such irregular collagen turnover compromises vascular stability and increases the risk of thrombus formation. Furthermore, inflammation induced by ER stress may create a pro-thrombotic state, heightening the risk of developing DVT [4,6].

#### 3.1.3. The Role of HSP47 in Procollagen Interaction

The role of HSP47 in maintaining cellular homeostasis and endoplasmic reticulum (ER) quality control is pivotal, particularly in the context of procollagen maturation and the secretory pathway. HSP47 acts as a specific chaperone for procollagen, ensuring proper protein folding and preventing aggregation and toxicity that arise from misfolded proteins, similar to general chaperones in protein homeostasis. It collaborates with other ER-resident proteins, such as Prolyl 4-Hydroxylase and protein disulfide isomerase, to facilitate collagen maturation through oxidative folding and essential post-translational modifications. HSP47 also plays a crucial role in retaining misfolded procollagen in the ER, preventing the secretion of non-functional proteins, thereby safeguarding the quality of proteins entering the extracellular environment [1,16,21].

Additionally, HSP47 is integral to the unfolded protein response (UPR), a protective mechanism activated when the ER is overwhelmed by misfolded proteins. This chaperone helps in upregulating other chaperones and managing the protein load through degradation pathways, reinforcing its crucial position in quality control and stress response. The functionality of HSP47 is directly linked to cellular health; its dysfunction can lead to chronic ER stress, contributing to various pathologies such as fibrosis, connective tissue disorders, and cardiovascular diseases. Therefore, HSP47 not only supports protein folding and quality control within the ER but also plays a significant role in maintaining cellular homeostasis and preventing disease through effective protein management [1,4,22].

#### 3.1.4. Procollagen Stability and the Role of HSP47

HSP47 serves as a critical collagen-specific chaperone in the endoplasmic reticulum (ER), crucial for ensuring the proper folding and stabilization of procollagen. Unlike other chaperones such as Protein Disulfide Isomerase (PDI) and Prolyl 4-Hydroxylase (P4H), which assist in oxidative folding and post-translational modifications, HSP47 directly binds procollagen during its folding process. This interaction is essential for preventing premature aggregation and facilitates the correct formation of the stable triple-helix structure of collagen, thereby influencing the quality of collagen that cells secrete [23,24,25].

Collagen, being a fundamental component of connective tissues, plays a key role in maintaining the structural integrity and strength of organs, skin, cartilage, tendons, and blood vessels. Dysfunction in HSP47 can lead to misfolded or unstable procollagen, adversely affecting collagen formation and consequently weakening connective tissue architecture. This dysfunction is linked to various connective tissue disorders (cTDs), such as osteogenesis imperfecta, Ehlers–Danlos syndrome, and fibrosis. These disorders often result from mutations in collagen or HSP47, leading to clinical symptoms such as fragile bones, hyperelastic skin, and impaired wound healing due to improperly assembled collagen structures [26,27,28].

The critical role of HSP47 in maintaining procollagen stability underscores its potential pathophysiological significance in connective tissue disorders. It highlights the promise of targeting HSP47 in therapeutic strategies aimed at optimizing collagen stability and addressing extreme connective tissue pathway disorders [1,4,29].

#### 3.1.5. Collagen and Connective Tissue Disorders

Collagens are essential components of the extracellular matrix, playing a critical role in tissue structure and function. The collagen-specific chaperone HSP47 is integral to collagen biosynthesis, particularly in stabilizing and assembling various collagen types, which has significant implications for human collagenopathies [1,25,30].

Notably, HSP47 directly affects collagen biosynthesis and is central in connective tissue disorders (osteogenesis imperfecta and Ehlers–Danlos syndrome). HSP47 expression levels vary in diseases where collagen structures are mutated or collagen disposal is deregulatory. In addition, HSP47 has distinct interactions with different collagen subtypes, including transmembrane collagens such as Collagen XVII and IX. Overexpression of these collagens aids in providing stability to the tissue and repair mechanisms including wound healing and fibrosis-related remodeling. HSP47 interacts differently with various collagen subtypes, essential for proper tissue function. It primarily associates with fibrillar collagens (types I, II, III), crucial for the structural stability of bone, cartilage, and skin. Its role in these interactions ensures the correct triple-helix formation needed for mechanical strength. Conversely, HSP47’s interaction with basement membrane collagens (types IV, VII) involves different binding kinetics, crucial for the structural integrity of tissues like skin and blood vessels [1,28,31].

As a chaperone, HSP47 aids in procollagen folding, identifying, and retaining misfolded procollagen within the ER to maintain protein quality. It also facilitates the transport of correctly folded procollagen to the Golgi apparatus for further processing before secretion into the extracellular matrix [1,14,32] (Table 2).

In multiple studies evaluating the function of HSP47, knockout models have been used to explore the effects of the loss of HSP47 on collagen stability. For example, one prominent study analyzed collagen composition in HSP47-deficient mice by mass spectrometry, revealing impaired collagen integrity.

HSP47 is a key player in collagen biosynthesis, directly affecting the stability and assembly of collagen subtypes. Its role in maintaining collagen integrity has profound implications for connective tissue disorders, offering novel insights for therapeutic targets in treating such pathologies [27,33].

#### 3.1.6. Binding Characteristics of HSP47

HSP47, a collagen-specific chaperone localized in the endoplasmic reticulum (ER), plays a critical role in the synthesis and folding of procollagen, significantly influencing structural integrity and disease prevention in connective tissues. The specific binding interactions between HSP47 and various collagen types are crucial to understanding tissue health and the pathophysiology of collagen-related diseases [1,34,35].

As a structural protein, HSP47 promotes the stable formation of the triple-helix structure in procollagen, which is essential for maintaining the mechanical strength and elasticity of collagen matrices in tissues such as skin, bone, and cartilage. By preventing the aggregation and misfolding of procollagen, HSP47 serves a protective function against diseases caused by collagen dysregulation, such as osteogenesis imperfecta and Ehlers–Danlos syndrome. Efficient binding of HSP47 minimizes the production of incorrectly synthesized collagen precursors, thereby mitigating the onset of associated pathologies [27,28,36].

#### 3.1.7. The Importance of Collagen in Atherosclerotic Plaques

HSP47’s role in collagen dynamics modulates stability of atherosclerotic plaque. Research indicates the potential role of HSP47 modulation in promoting fibrin plaque stability, thereby mitigating the risk of rupture and ultimately decreasing the incidence of cardiovascular events [4,37,38].

The relationship between collagen thickness in atherosclerotic plaques and the likelihood of plaque rupture is a significant topic in cardiovascular health research. Understanding how collagen content and structure contribute to plaque stability offers valuable insights into risk assessment and therapeutic strategies [39,40].

Early research established a crucial link between collagen content and plaque stability. The study demonstrated that plaques with thicker fibrous caps, which contained a higher amount of collagen, exhibited greater structural stability. Specifically, collagen-dominant caps were associated with a reduction in rupture risk compared to caps consisting primarily of thin collagen layers. Further investigation reported an inverse correlation between collagen content in the fibrous cap and the risk of rupture. The study measured collagen thickness in human carotid plaques, revealing that plaques with a thickness of less than 65 μm were significantly more prone to rupture [41,42,43].

Another study reinforced the significance of collagen dynamics by showing that lower collagen content in atherosclerotic plaques was associated with heightened inflammatory signals. This inflammation correlated with increased propensity for rupture. The study emphasized that a favorable collagen-to-fat ratio was linked to fewer rupture events, suggesting that maintaining a robust collagen matrix is vital for plaque stability [44,45,46].

Advanced imaging modalities, such as magnetic resonance imaging (MRI) and computed tomography (CT), are crucial for non-invasive assessments of plaque structure, including collagen content and thickness. These technologies enable precise categorization of patients based on their risk for cardiovascular events, facilitating early intervention and management strategies [47].

Current research also explores biomarkers associated with collagen turnover, specifically degradation products like C-telopeptides. These biomarkers have the potential to predict plaque stability and risk of rupture, thereby enhancing risk stratification in clinical settings [48,49].

Emerging therapeutic strategies aim to target collagen metabolism to stabilize plaques. Experimental approaches include increasing collagen production via specific growth factors or anti-inflammatory agents to promote a stable collagenous matrix. Additionally, investigational studies are evaluating therapies that enhance collagen crosslinking or inhibit collagenase activity with the intent to strengthen plaque integrity and reduce the risk of rupture. Collectively, these studies and perspectives provide a cohesive framework linking measurable changes in collagen thickness to plaque rupture risk. Understanding the dynamics of collagen in atherosclerotic plaques not only aids in diagnosing cardiovascular risk but also opens avenues for targeted therapeutic interventions aimed at stabilizing plaques and preventing cardiovascular events. As research advances, focusing on collagen dynamics may yield transformative strategies for managing cardiovascular health [50,51].

#### 3.1.8. The Backbone of Atherosclerotic Plaques

The prospect of targeting heat shock protein 47 presents significant therapeutic potential for modifying plaque stability and progression in atherosclerosis. This review examines the potential advantages of HSP47 targeting and its implications for cardiovascular health. HSP47 functions as a critical molecular chaperone responsible for collagen biosynthesis and stability. By modulating HSP47 activity, it may be possible to enhance the maturation of collagen within existing atherosclerotic plaques. Increased collagen stability could lead to thicker fibrous caps, thereby reducing the likelihood of plaque rupture. Evidence suggests that strengthening collagen networks correlates with more stable plaques, which in turn is associated with a lower incidence of acute cardiovascular events [4,5,37].

The inhibition of HSP47 may influence collagen dynamics within plaques by promoting collagen degradation and reducing aberrant collagen accumulation. This process could lead to a decrease in plaque lumen narrowing and contribute to the formation of more quiescent, less inflammatory lesions over time. HSP47 plays a role in cellular responses to stress and is implicated in inflammatory processes within plaques. Targeting HSP47 may alleviate inflammation, which is known to contribute to plaque instability. By reducing inflammatory cell infiltration and cytokine release, HSP47 modulation may create a more favorable microenvironment for plaque stabilization.

There is potential for developing small-molecule inhibitors or monoclonal antibodies aimed at HSP47 as direct therapeutic agents. These agents could be designed to specifically alter HSP47’s function, thereby influencing extracellular matrix (ECM) remodeling pathways during the progression of atherosclerosis.

Further research is essential to elucidate the biological role of HSP47 in atherosclerotic plaques and to establish the safety and efficacy of HSP47-targeted therapies in large-scale clinical trials. While preclinical studies show promise in modulating HSP47 expression and its effects on plaque composition, human clinical trials are necessary to validate these findings.

Targeting HSP47 represents a promising strategy for enhancing plaque stability and mitigating progression in atherosclerosis. By addressing these alterations to plaque stability, therapies aimed at HSP47 could offer new avenues for cardiovascular protection and reduce the risk of acute events such as myocardial infarctions (MI) and strokes. Continued investigation into the therapeutic potential of HSP47 will be crucial for translating these findings into clinical practice [4,52,53].

#### 3.1.9. HSP47: Targeting Plaque Stability in Atherosclerosis

Recent research highlights the potential of targeting heat shock protein 47 (HSP47) in managing atherosclerosis and enhancing plaque stability. Experimental approaches, including histological analyses, immunohistochemistry, and mouse models, reveal the significant role of HSP47 in collagen metabolism and plaque integrity. Studies demonstrate that HSP47 expression correlates with collagen organization in atherosclerotic plaques, while genetic modifications in mouse models illustrate impaired collagen handling and decreased plaque stability when HSP47 is deficient.

Furthermore, in vitro assays using vascular smooth muscle cells and macrophages support the influence of HSP47 on collagen synthesis and inflammatory processes. The implications of HSP47 research extend to therapeutic strategies, as understanding its modulation of collagen turnover may guide the development of targeted treatments aimed at stabilizing plaques and preventing rupture.

Incorporating HSP47 targeting into personalized medicine may enhance cardiovascular outcomes by tailoring interventions based on individual plaque characteristics. Additionally, combining HSP47-targeted therapies with established treatments like statins could yield synergistic benefits in reducing cardiovascular risk.

Overall, targeting HSP47 offers promising avenues for therapeutic development in atherosclerosis treatment, warranting further exploration to solidify its role as a critical target in reducing the burden of atherosclerotic disease [4,37,53,54].

### 3.2. Expression Patterns of HSP47 in Atherosclerosis

Transforming Growth Factor Beta 1 (TGF-β1) and Fibroblast Growth Factor 2 (FGF-2) are key cytokines that regulate heat shock protein 47 (HSP47) and collagen production, essential for extracellular matrix (ECM) maintenance in various tissues, including atherosclerotic plaques. TGF-β1 increases HSP47 expression and collagen synthesis through the SMAD signaling pathway, whereas FGF-2 enhances HSP47 and type I procollagen expression via the MAPK/ERK and PI3K/Akt pathways [55,56].

However, the interplay between HSP47 and type I procollagen can be disrupted under inflammatory conditions, oxidative stress, and hypoxia, leading to compromised collagen synthesis and plaque instability. For instance, high levels of TNF-α can upregulate HSP47 while downregulating type I procollagen expression, resulting in weak fibrous caps prone to rupture.

Therapeutically, modulating HSP47 expression presents a promising approach for stabilizing plaques. Activating HSP47 may enhance collagen deposition and plaque integrity, while inhibition may be beneficial in cases of excessive collagen accumulation. Additionally, HSP47 levels could serve as biomarkers for plaque stability, aiding in cardiovascular risk stratification. Combining HSP47-targeted therapies with existing treatments, such as statins and anti-inflammatory agents, could optimize outcomes by synergistically promoting collagen production and plaque stability.

Overall, understanding the regulatory role of TGF-β1 and FGF-2 on HSP47 highlights the potential for targeted therapeutic strategies in the prevention and management of atherosclerosis and its related cardiovascular events [4,37,55,56].

#### 3.2.1. Overview of Collagen Types and Their Functions

The externalization of phospholipids, particularly phosphatidylserine (PS), on activated platelets is essential for promoting coagulation, a critical step in hemostasis to prevent excessive bleeding following vascular injury. Upon injury, platelets adhere to collagen and extracellular matrix components, initiating activation and morphology changes-from disk-like shapes to sphere-like forms with filopodia extensions. This transformation increases the surface area available for interactions and exposes PS on the outer membrane leaflet, providing a pro-coagulant surface [57,58,59].

Exposed PS enhances the binding affinities of key coagulation factors, such as factor X and prothrombin, facilitating thrombin generation and promoting further platelet aggregation. This interaction orchestrates the formation of catalytic complexes necessary for converting prothrombin into thrombin and fibrinogen into fibrin, effectively driving the coagulation cascade [57,59,60].

Tissue factor, expressed at the site of injury, interacts with factor VIIa, forming a complex that activates factor X and further enhances coagulation through its localized activity on phospholipid-rich platelets. This localized concentration of active factors amplifies the coagulation response, increasing thrombin production and fibrin generation, with calcium ions (Ca^2+^) playing a crucial role in facilitating factor binding to the phospholipid surfaces [61,62].

Overall, the exposure of phospholipids on activated platelets significantly enhances the binding and activity of various coagulation factors, promotes catalytic complex formation, and amplifies the coagulation cascade. Understanding these mechanisms is vital for developing therapeutic strategies to manage conditions of excessive hemorrhage or thrombosis effectively [57,59,63].

#### 3.2.2. Role of Collagen in Hemostasis

Type IV collagen plays a pivotal role in hemostatic dynamics by regulating coagulation factors such as factor IX (FIX) and von Willebrand factor (VWF). FIX, a vitamin K-dependent serine protease, directly binds to type IV collagen through its propeptide region, which is essential for localizing FIX to vascular injury sites and facilitating its activation by factor XIIa or the tissue factor VIIa complex. Studies indicate that immobilized collagen type IV enhances FIX’s binding affinity, significantly improving its activation rate compared to other surfaces. VWF, an essential mediator of hemostasis, binds to collagen type IV, which helps anchor VWF at injury sites for platelet adhesion. This interaction is crucial under high shear stress conditions, promoting the formation of a primary hemostatic plug by enabling VWF to bind platelet glycoprotein Ib (GPIb). The presence of type IV collagen stabilizes thrombus formation, as it enhances platelet activation and aggregation through VWF interactions. In vitro studies reveal that the binding of FIX to collagen type IV markedly increases FIX activation rates, while animal models show that collagen IV supplementation can partially restore platelet function in VWF knockout mice, highlighting its stabilizing effects in hemostasis. Understanding the interactions between type IV collagen, FIX, and VWF is vital for developing therapeutic strategies aimed at enhancing hemostatic potential in patients with bleeding disorders. Overall, these mechanisms underscore the importance of type IV collagen in promoting effective hemostasis, particularly following vascular injury [64,65,66].

#### 3.2.3. Platelet Adhesion and Shear Rate Dynamics

Single-layer adhesion and aggregation are two critical processes in platelet function during hemostasis. Single-layer adhesion refers to the initial attachment of platelets to exposed extracellular matrix (ECM) components, such as collagen and von Willebrand factor (VWF), at sites of vascular injury. This process involves interactions between specific platelet surface receptors (e.g., glycoprotein Ib/IX/V with VWF and integrin α2β1 with collagen), resulting in the formation of a monolayer of platelets that aids in blocking blood loss and establishing the initial hemostatic plug.

Aggregation, on the other hand, occurs after initial adhesion, where activated platelets recruit and aggregate into a larger mass or thrombus. This process is driven by the release of soluble agonists (e.g., ADP and thromboxane A2) and conformational changes in integrins (e.g., αIIbβ3), enabling platelet–platelet interactions facilitated by fibrinogen bridging. The resulting platelet-rich thrombus covers tissue damage and prevents bleeding. The dynamics of blood flow shear rates are crucial in influencing these processes. At high shear rates, such as those in arterioles, platelet adhesion occurs rapidly, but aggregation may be limited due to strong binding forces that can disperse platelets before full aggregation. Understanding shear dynamics is essential for assessing thrombus formation efficiency and risk stratification for cardiovascular events, helping guide therapeutic interventions such as antithrombotic treatments.

In summary, single-layer adhesion represents the initial tethering of platelets to injury sites, while aggregation involves the clustering of platelets into a stable thrombus. Investigating shear rate dynamics is vital for optimizing thrombus formation and developing appropriate clinical strategies to mitigate the risks of thrombosis and bleeding [67,68,69].

#### 3.2.4. Characterization of Collagen Fragments and Platelet Response

Research on linear and triple-helical peptides presents promising avenues for therapeutic strategies targeting coagulation and collagen-related disorders.

Targeting platelet function: linear peptides derived from collagen sequences have been shown to influence platelet adhesion and activation. These peptides can inhibit platelet aggregation by blocking binding sites on glycoprotein receptors, paving the way for the development of specific antiplatelet therapies aimed at preventing thrombus formation in conditions like atherosclerosis and minimizing re-thrombosis during surgical procedures.

Increased stability of collagen: triple-helical peptides that mimic the structural stability of native collagen may enhance vascular collagen stabilization, potentially preventing aneurysms. Their application can support tissue repair and reinforce collagen networks within atherosclerotic plaques, reducing rupture risk and lowering the incidence of cardiovascular events.

Drug delivery systems: the distinctive structural features of linear and triple-helical peptides can be harnessed for drug delivery systems. Peptide carriers designed to target collagens or adhesion molecules could facilitate the localized delivery of therapeutic agents to vascular injury sites or areas with abnormal collagen deposition.

Biomarker development: investigating collagen-derived peptides may help identify biomarkers indicative of pathophysiological conditions related to disturbed collagen metabolism, such as fibrotic diseases or vascular complications. Monitoring circulating peptide concentrations could provide valuable diagnostic and prognostic insights [70].

Responsive personalized medicine: insights into the interactions between linear and triple-helical peptides and platelet receptors can inform personalized medicine approaches. Understanding polymorphisms related to peptide interactions can guide the selection of optimal treatment regimens, maximizing therapeutic efficacy while minimizing side effects.

Findings from studies on linear and triple-helical peptides highlight new therapeutic strategies for modulating platelet function, enhancing tissue repair, and optimizing vascular health. These developments lay the groundwork for targeted therapies addressing various vascular and coagulation abnormalities, advancing the potential for personalized medicine in this field [71,72,73,74].

#### 3.2.5. Interaction Between Integrins and Collagen

Integrin α2β1 acts as a crucial collagen receptor on platelets, significantly impacting platelet adhesion and aggregation during hemostasis. Its modulation presents several therapeutic opportunities, particularly in cardiovascular diseases and thrombotic disorders.

Antithrombotic strategies: inhibition of α2β1 may serve as a novel antithrombotic approach, especially for patients at high risk for thrombus formation due to atherosclerosis or vascular injury. By reducing platelet adhesion to collagen, this inhibition could decrease thrombus growth and lower the incidence of acute cardiovascular events such as myocardial infarction and stroke.

Enhancing platelet function: conversely, activating α2β1 is beneficial in contexts like bleeding disorders or surgical settings, as it promotes effective platelet adhesion to collagen-rich tissues, enhancing hemostatic efficiency.

Collagen dynamics regulation: modulating α2β1 can also affect platelet adhesion in collagen-rich ECM, helping to restore proper collagen dynamics during disease states associated with excessive or insufficient remodeling, contributing to improved healing post-injury.

Personalized therapies: patient-specific variations in α2β1 expression may influence therapeutic responses, enabling the development of customized treatment strategies tailored to individual platelet receptor profiles or genetic backgrounds.

PDI’s role in α2β1 regulation: protein disulfide isomerase (PDI) enhances α2β1 activation by facilitating conformational changes that shift the integrin to a high-affinity state for collagen. PDI’s regulation intersects with other platelet activation pathways, including GPVI and thrombin signaling, which may amplify platelet responses and increase the risk of thrombosis in pathological conditions like atherosclerosis.

The ability to modulate α2β1 activity presents significant therapeutic potential in managing hemostatic responses and thrombotic risks. Understanding the regulatory role of PDI and the interplay of various activation pathways will be crucial for developing targeted therapies that balance effective hemostasis without triggering excessive thrombosis [75,76,77].

#### 3.2.6. The Implications of GPVI and Integrin Interplay

Glycoprotein VI (GPVI) and integrin α2β1 are critical receptors on platelets that mediate their activation and adhesion to collagen during hemostasis. These receptors exhibit both redundancy and synergy in their functions, ensuring effective platelet responses to vascular injury.

Redundancy in function: both GPVI and α2β1 bind collagen, a key ligand present in the extracellular matrix upon vascular damage. Their overlapping functions allow compensation if either receptor pathway is impaired, enabling continued platelet adhesion and activation. GPVI primarily activates platelets through immunoreceptor tyrosine-based activation motif (ITAM) signaling, leading to calcium mobilization, while α2β1 enhances activation via integrin signaling pathways.

Synergy in function: at intermediate collagen concentrations, GPVI and α2β1 work cooperatively. Initial activation through GPVI enhances the affinity of α2β1 to collagen, promoting stable platelet attachment and larger thrombus formation, particularly in high-shear environments like arterioles.

Clinical significance: understanding the interactions between GPVI and α2β1 is crucial for developing antiplatelet therapies. Targeting GPVI can reduce platelet reactivity; however, α2β1 may still function independently, necessitating a dual inhibition strategy for more effective antithrombotic effects. Moreover, inhibiting these pathways can come with the risk of increased bleeding, emphasizing the need for careful dosing and individualized treatment strategies.

The redundancy and synergy between GPVI and α2β1 underpin the complexity of hemostatic responses and have important implications for antiplatelet therapy development. Insights into these interactions can lead to improved therapeutic strategies for managing thrombotic diseases while minimizing side effects [70,78].

#### 3.2.7. Novel Investigations into Platelet Collagen Receptors

Single-chain antibodies (scFvs) are engineered antibodies formed by linking the variable regions of heavy and light chains with a flexible peptide, facilitating access to conformational epitopes. Their small size, ease of production, and specificity make scFvs valuable tools for research and therapeutic applications, particularly in targeting receptors involved in coagulation, such as glycoprotein VI (GPVI).

Development and validation: the development of scFvs commonly employs phage display technology, allowing the creation of diverse libraries and the selection of high-affinity binders against target antigens. Validation of scFvs includes techniques like ELISA and flow cytometry to confirm binding and biological activity.

Species-specific differences in collagen receptors: research highlights the functional implications of species-specific variations in collagen receptors like GPVI and integrin α2β1. Studies indicate that human GPVI shows higher binding affinity to type I collagen compared to murine GPVI, affecting therapeutic efficacy and safety when translating findings from animal models to human applications.

Impact of GPVI polymorphisms on cardiovascular risk: genetic studies have linked polymorphisms in the GP6 gene, which encodes GPVI, to altered platelet function and increased cardiovascular disease risk. Specifically, certain GPVI polymorphisms correlate with heightened platelet reactivity to collagen, leading to an increased incidence of myocardial infarction. These genetic variations may influence platelet activation and inflammatory cytokine release, thereby contributing to atherogenesis.

ScFvs offer a promising platform for therapeutic development, while understanding interspecies differences in collagen receptor function and the implications of GPVI polymorphisms can enhance personalized medicine approaches in cardiovascular care. Genotyping GPVI variations may aid in defining individual cardiovascular risk and in tailoring effective anti-platelet therapies [79,80,81].

#### 3.2.8. The Importance of Adhesion Motifs and Cytoskeletal Extension

Platelet adhesion is a crucial process involving interactions between platelet receptors, such as integrin α2β1, GPVI, and CD36, with collagen fibers, significantly influenced by shear stress environments. Understanding these interactions is vital for addressing thrombotic diseases, including arterial stenosis and venous thrombosis.

Effective platelet adhesion and cytoskeletal extension are essential for thrombus formation. Under high shear conditions, robust receptor engagement, particularly integrin α2β1 and the von Willebrand factor (VWF) axis, is necessary for stable adhesion. Consequently, antiplatelet therapies may need to be tailored based on the hemodynamic environment-different approaches may be required for patients experiencing high shear stress versus those in slower flow conditions.

The hypothesis of a two-site, two-step adhesion mechanism reveals that initial weak binding via GPVI is followed by stronger adhesion through integrin α2β1. This model highlights the varied roles of platelet receptors in different thrombotic contexts, where GPVI and CD36 dominate in venous thrombus formation, promoting aggregation in low-flow conditions.

In arterial stenosis, elevated shear rates create a pro-thrombotic environment that increases platelet activation and underscores the need for aggressive antithrombotic therapies to prevent cardiovascular events. In the case of venous thrombosis, pronounced receptor interactions can lead to large, fibrin-rich thrombi obstructing venous flow [82].

Additionally, the presence of long, insoluble collagen fibers in pathological conditions can enhance thrombus stability, indicating that disrupting receptor–collagen interactions might be a viable therapeutic strategy.

Insights into platelet receptor interactions with collagen under varying shear conditions are critical for developing effective clinical approaches to manage thrombotic diseases. A deeper understanding of these dynamics will aid in the design of targeted antithrombotic therapies, ultimately improving patient outcomes in conditions characterized by abnormal thrombus formation [68,69,70,76,83].

#### 3.2.9. The Role of Collagen Type I in Thrombus Formation

Perfusion chambers are essential experimental tools that simulate physiological and pathophysiological conditions relevant to hemostasis and thrombosis, allowing researchers to investigate platelet behavior under controlled flow rates and collagen preparations.

Flow rates and shear stress: perfusion systems can replicate low and high shear rates, simulating venous and arterial blood flow conditions, respectively. This enables the study of the effects of shear stress on platelet adhesion and thrombus formation, providing insights into the dynamics of thrombus stability during vascular injury, such as in atherosclerosis or following angioplasty.

Collagen preparation: collagen is derived from human or animal tissues, providing control over the types and concentrations of collagen that mimic pathological conditions. The incorporation of other extracellular matrix components like fibronectin and von Willebrand factor (vWF) enhances the relevance of the models to real-world biological interactions.

Therapeutic targeting of vWF: vWF plays a critical role in platelet adhesion to collagen and stabilizes factor VIII in circulation, thus facilitating effective clotting. Under high shear conditions, vWF adopts an active form that promotes thrombus formation. Targeting vWF with antibodies or small molecules can reduce platelet aggregation, offering potential strategies for managing acute coronary syndromes and preventing thrombus formation following angioplasty.

Relevance to atherosclerosis and restenosis: understanding the interplay between vWF, platelets, and collagen is crucial in the context of atherosclerosis and post-angioplasty restenosis, where platelet activation can lead to adverse thrombotic events. Inhibiting vWF may represent a therapeutic avenue to prevent thrombus formation on ruptured plaques and reduce re-occlusion risks after stenting procedures.

Perfusion chamber studies provide valuable in vitro insights into platelet dynamics across varying shear conditions, highlighting their significance in developing antithrombotic therapies aimed at mitigating thrombotic complications associated with vascular diseases. This research trajectory promises to enhance clinical management and improve outcomes for patients at risk for thrombotic events [84,85,86,87].

### 3.3. Impact of HSP47 on Platelet Binding to Collagen

HSP47 plays a prominent role in platelet activity with impact on thrombus generation in hemostasis. It helps create stable collagen in the ECM and enables proper platelet–collagen interaction necessary for stable thrombus. HSP47 has been implicated in the development of both arterial and venous thrombi and has also been shown to inhibit thrombus formation during in vitro assays; as such, it represents a promising target for development of therapeutics in the setting of thrombotic events.

Targeting heat shock protein 47 (HSP47) emerges as a critical strategy for understanding collagen-mediated platelet interactions, particularly regarding the selective impact of HSP47 inhibitors. These inhibitors effectively reduce collagen-associated platelet adhesion and aggregation while leaving thrombin-induced activation unaffected, indicating HSP47’s specific role in stabilizing collagen–platelet interactions.

HSP47 inhibitors specifically influence collagen-mediated interactions by stabilizing collagen fibers and facilitating receptor engagement on platelets, primarily through receptors such as GPVI and integrin α2β1. In contrast, thrombin activates platelets through distinct receptors like PAR-1 and PAR-4, which do not rely on collagen binding or HSP47 stabilization, thereby remaining unaffected by HSP47 blockade.

The ability of HSP47 inhibitors to diminish platelet adhesion exclusively in the presence of collagen highlights HSP47’s role as a major modulator in the initial interactions between platelets and collagen. This direct involvement underscores HSP47’s importance in mediating platelet adhesion and the subsequent signaling necessary for thrombus formation.

Given its targeted influence on collagen-dependent platelet interactions, HSP47 presents a promising therapeutic target for preventing thrombosis without the side effects associated with broader antiplatelet therapies that disrupt ADP or thrombin signaling. Overall, HSP47 is best characterized as a primary mediator of collagen–platelet interactions, making it an attractive candidate for selective blockade in thrombotic disease management [6,52,88].

#### 3.3.1. The Effect of HSP47 Inhibition on Platelet Aggregation

The use of 96-well plate assays to examine the effects of heat shock protein 47 (HSP47) inhibition on platelet aggregation provides valuable insights into therapeutic strategies aimed at modulating thrombotic diseases.

These assays demonstrate high sensitivity in detecting collagen-mediated changes in platelet aggregation, allowing for both partial and complete inhibition depending on the collagen concentration. Assay reproducibility is ensured through multiple runs and the inclusion of appropriate controls, such as baseline wells devoid of inhibitors and standardized platelet preparations, to mitigate variability.

The results reveal a dose-dependent relationship in the inhibition of platelet aggregation by HSP47 inhibitors, indicating that complete inhibition occurs at lower collagen concentrations, while higher concentrations lead to variable inhibition levels. This suggests that competition between collagen and inhibitors at binding sites informs the effectiveness of HSP47 stabilization on platelet binding.

Given the critical role of HSP47 in platelet–collagen interactions, targeting this pathway may provide new strategies for managing thrombotic conditions, such as deep vein thrombosis and myocardial infarction. Optimizing inhibitor dosages can facilitate tailored therapeutic approaches that effectively reduce pathological platelet activity while preserving essential hemostatic functions.

The 96-well assay methodology offers a robust framework for assessing the potential of HSP47 inhibitors in platelet aggregation, contributing to the development of refined therapeutic strategies to combat thrombotic diseases through precise dose management [6,52].

#### 3.3.2. HSP47’s Role in Calcium Mobilization and Platelet Function

Inhibition of heat shock protein 47 (HSP47) has been shown to reduce calcium ion (Ca^2+^) mobilization, a critical process that influences platelet activation and thrombus stabilization. Ca^2+^ signaling is essential for platelet shape change, granule release, and aggregation, all of which are necessary for effective hemostasis and stable thrombus formation. HSP47 inhibition decreases Ca^2+^ rise in response to collagen stimulation, thereby preventing platelet activation and potentially destabilizing thrombi during thromboembolic events.

Reliable data on Ca^2+^ mobilization necessitate proper experimental controls, employing isotype control antibodies to establish the specificity of HSP47’s effects. Additional stimulants like thrombin and ADP help confirm that HSP47 inhibition specifically impacts collagen signaling rather than universally compromising platelet responses. Notably, while the response to CRP-XL (a GPVI-activating agent) is delayed with HSP47 inhibition, it is not entirely abolished, indicating that HSP47 is significant for collagen-mediated interactions but not essential for all platelet activation pathways.

Targeting HSP47 presents beneficial therapeutic implications for thrombotic disorders, such as deep vein thrombosis and acute coronary syndrome. Selective inhibition of platelet aggregation could be achieved while maintaining necessary hemostatic functions. Furthermore, investigating individual variability in responses to HSP47 inhibition may lead to personalized treatments that enhance efficacy and minimize bleeding risks.

HSP47 plays a crucial role in Ca^2+^ mobilization vital for platelet function and thrombus stability. The findings underscore the potential for developing tailored therapies for thrombotic conditions, facilitated by robust experimental controls in research [6,52].

#### 3.3.3. Influences of HSP47 on Thrombus Formation and Aggregation

To investigate HSP47’s role in platelet reactivity under physiological conditions, DiOC6-labeled blood from mice lacking platelet HSP47 and control mice was perfused over collagen. Thrombus formation, measured by fluorescence intensity, was reduced by about half in HSP47-deficient mice compared to controls. Similarly, human blood treated with HSP47 inhibitors showed a reduction in thrombus volume by roughly one-third with SMIH and a slightly smaller reduction with anti-HSP47 compared to controls.

Importantly the role of HSP47 in collagen stability has been demonstrated in murine models and corroborated in humans. While HSP47 deficiency leads to severe phenotypes in mice, including embryonic lethality because of unstable collagen type I, issues of collagen stability directly relate to HSP47. Interestingly, high HSP47 levels in adipose tissues are linked with obesity and increased collagen deposition in humans, implying a similar functional significance for mammals. The conservation of this function throughout species highlights the importance of HSP47 as a key player in the interplay between collagen and health [6,33].

#### 3.3.4. Evaluating the Specificity of HSP47 in Platelet Interactions

Heat shock protein 47 (HSP47) is a molecular chaperone with a high affinity for collagen, playing a crucial role in the synthesis and functionality of different collagen types. Its specificity in binding to procollagen and mature collagen fibrils is determined by the unique structural and biochemical properties of the binding site, particularly interactions with proline and hydroxyproline residues essential for collagen’s triple-helical conformation.

Unlike most serpins, HSP47 does not serve as a protease inhibitor; instead, it stabilizes collagen by engaging with its hydrophobic zones and forming salt bridges with specific amino acids, such as arginine and aspartate. HSP47 exhibits promiscuous binding to various collagen types, including I, II, III, and IV, and preferentially interacts with higher-order collagen structures containing interrupted triple helices, ensuring proper collagen folding and assembly.

Moreover, HSP47 is critical for platelet adhesion and aggregation by stabilizing collagen in the extracellular matrix. Inhibiting HSP47 leads to decreased platelet aggregation and thrombus formation specifically in response to collagen while not affecting pathways activated by thrombin or GPVI.

The binding specificity of HSP47 highlights its essential role in collagen dynamics and platelet function, suggesting it as a promising therapeutic target for disorders related to connective tissue and thrombotic diseases. Understanding these mechanisms could pave the way for the development of new treatments aimed at modulating collagen interactions and platelet recruitment [5,6,15].

#### 3.3.5. Exploring Early Signaling Events Following GPVI Ligation

HSP47 plays a critical role in binding distinct types of collagen, which significantly influences the regulation of platelet activation and aggregation. This relationship offers valuable insights into the mechanisms that maintain hemostatic balance and may lead to the development of novel therapeutic modalities.

During vascular injury, exposed collagen serves as a potent inducer of platelet activation, while HSP47 stabilizes collagen structure within the extracellular matrix. The interaction between platelets and collagen is vital for activating signaling pathways via GPVI and integrin α2β1, resulting in platelet adhesion, shape change, and degranulation. HSP47’s role in structurally stabilizing collagen enhances platelet aggregation by facilitating the optimal clustering of adhered platelets to form a stable thrombus [80].

In contrast, the inhibition of HSP47 significantly reduces platelet aggregation and impairs crucial downstream signaling pathways, including calcium mobilization, which is essential for optimal platelet function. This understanding of HSP47’s functional role reveals potential therapeutic targets for ameliorating hemostatic disorders characterized by either excessive or insufficient platelet aggregation. Specifically, HSP47 modulation could provide a new therapeutic avenue for preventing abnormal platelet coalescence during acute events, such as myocardial infarction.

Moreover, the SH2 domain of HSP47 is integral to its binding with collagen and subsequent platelet activation. These insights may help inform novel therapeutic strategies aimed at enhancing patient outcomes in bleeding disorders and vascular diseases. Overall, targeting HSP47 could be instrumental in developing treatments that balance hemostatic function and mitigate thrombotic risk [6,52,88].

#### 3.3.6. Intravital Microscopy: Examining Platelet Recruitment Post-Injury

HSP47’s specificity for binding various collagen types positions it as a critical regulator in platelet activation and aggregation while maintaining a delicate balance between hemostasis and thrombosis. This has significant implications for therapeutic strategies targeting cardiovascular diseases.

Platelet activation is primarily initiated by the exposure of collagen following vascular injury. HSP47 enhances the structural stability of collagen in the extracellular matrix, facilitating effective receptor binding and subsequent platelet signaling through receptors such as glycoprotein VI (GPVI) and integrin α2β1. These interactions govern platelet adhesion and trigger essential processes, including shape change and granule secretion. HSP47’s stabilization of collagen allows for effective platelet aggregation, promoting thrombus formation.

Inhibition of HSP47 markedly decreases platelet aggregation and disrupts downstream signaling, particularly calcium mobilization, which is vital for platelet function. An established model for studying vascular responses to collagen enables real-time imaging of thrombus dynamics in responses to injury, showcasing how HSP47’s role affects thrombus growth compared to other injury models.

While early thrombus kinetics remain similar across models, HSP47 inhibition leads to reduced growth phases, highlighting its significance in thrombus development. Targeting HSP47 may present a novel therapeutic approach to selectively modulate thrombus stability and size, potentially preventing excessive aggregation during acute conditions like myocardial infarction while still allowing necessary platelet–collagen adhesion for hemostasis.

HSP47’s binding specificity to collagen is crucial for efficient platelet activation and aggregation. Understanding its role opens avenues for developing targeted therapies that can better manage hemostatic disorders associated with dysfunctional platelet aggregation, offering potential solutions for improving vascular health and controlling thrombotic conditions [6,52,88].

#### 3.3.7. Impact of HSP47 on Hemostasis in Mice

Inhibition of heat shock protein 47 (HSP47) using the small molecule inhibitor SMIH, alongside data from platelet-specific HSP47-deficient mice, highlights HSP47 as a critical regulator of hemostasis. The administration of SMIH to C57Bl/6 mice resulted in an increase in tail bleeding time, indicative of hemostatic dysfunction and impaired platelet function. Further analysis revealed that HSP47-deficient mice exhibited a 1.8-fold increase in bleeding time compared to controls, confirming the essential role of HSP47 in maintaining normal platelet activity.

These findings correlate with clinical observations in humans, where patients with functional platelet disorders, such as Glanzmann thrombasthenia (characterized by defective integrin αIIbβ3), experience prolonged bleeding times due to impaired platelet aggregation. The parallels between enhanced bleeding resulting from HSP47 inhibition in experimental models and the challenges faced by patients with intrinsic platelet defects underscore HSP47’s vital function in preserving platelet interactions with collagen, primarily mediated by integrins.

While HSP47 presents a promising target for therapeutic interventions aimed at managing pathological thrombus formation, it is crucial to balance the need for sufficient hemostatic responses against the risk of excessive clotting. The extended bleeding times associated with HSP47 inhibition further emphasize the importance of platelet function in hemostatic control, providing a rationale for targeting HSP47 in the treatment of hemostatic disorders while minimizing the risk of bleeding complications. Overall, HSP47 stands out as a significant player in the regulation of platelet function and hemostasis [6].

#### 3.3.8. The Role of Surface Receptors in Platelet Activation

Platelet adhesion to collagen at sites of vascular injury is a vital step in hemostasis, primarily mediated by two key receptors: glycoprotein VI (GPVI) and integrin α2β1. The significant overlap between these receptors has profound implications for platelet physiology and the development of therapeutic strategies.

Glycoprotein VI (GPVI): this receptor is crucial for initiating signaling pathways that activate platelets and facilitate thrombus formation in response to collagen.

Integrin α2β1: serving as the primary collagen-binding site, α2β1 is essential for the firm adhesion of platelets to the extracellular matrix, enhancing platelet–collagen interactions.

In addition to GPVI and integrin α2β1, other receptors such as integrin αIIbβ3 and CD36 also contribute to collagen adhesion, increasing redundancy in platelet response mechanisms.

The redundancy among collagen receptors allows for compensatory signaling pathways when one receptor’s function is impaired. For instance, if GPVI signaling is diminished, integrin α2β1 can help rescue platelet adhesion, thus preserving hemostatic function.

Differences in receptor expression levels among individuals can influence platelet responses to collagen, indicating the need for a multifaceted approach in therapeutic targeting. Recognizing the redundancy in receptor functions is essential for developing effective treatment strategies for thromboembolic diseases.

Focusing solely on GPVI as a therapeutic target may lead to suboptimal outcomes due to compensatory mechanisms from other receptors. Therefore, a strategy that targets multiple receptors may enhance therapeutic efficacy while mitigating risks associated with thrombotic disorders.

Vascular conditions and shear stress can alter thrombus formation dynamics, necessitating the involvement of multiple receptors to maintain proper function under high-shear environments.

Overall, the redundancy of receptors mediating platelet–collagen adhesion is critical for maintaining hemostatic balance. This understanding provides a rationale for exploring receptor interactions further, paving the way for novel receptor-based therapies aimed at improving the management of bleeding and thrombotic disorders [78,82,89].

#### 3.3.9. Interaction of HSP47 with Collagen in Platelet Function

HSP47, a collagen-specific molecular chaperone, plays a critical role in the interactions between platelets and collagen during vascular injury, essential for thrombus stability and successful hemostasis. Following vascular damage and collagen exposure, platelets adhere through various receptors, including glycoprotein VI (GPVI) and integrin α2β1, forming a vital platelet plug to prevent bleeding. HSP47 enhances the binding between platelets and collagen after this initial adhesion, promoting stability in thrombus formation.

Recent studies utilizing flow cytometry and imaging have shown that HSP47 facilitates the recruitment of additional platelets to the thrombus, thereby strengthening platelet–collagen interactions. Inhibition of HSP47 leads to less stable platelet aggregates, underscoring its importance in maintaining thrombus integrity. Furthermore, HSP47’s ability to bind to phosphorylated forms of the extracellular matrix (ECM) and interact with various ECM proteins, including fibronectin, laminin, and proteoglycans, indicates its broader role in structural support and platelet activation.

These multifaceted functions of HSP47 highlight its significance in hemostasis by stabilizing platelet–collagen interactions and facilitating dense platelet aggregation. Understanding HSP47’s interactions with diverse ECM components may provide new insights into thrombus biology and uncover potential therapeutic strategies for managing hemostatic disorders and enhancing vascular health [6,90].

### 3.4. Understanding HSP47’s Modulatory Effects on Platelet Responses

Heat shock protein 47 (HSP47) is recognized for its essential role in thrombosis, particularly in platelet–collagen interactions that strengthen thrombus structure and facilitate efficient hemostasis. However, its relevance extends beyond thrombotic conditions to include critical functions in wound healing and tissue repair, underscoring its multifactorial importance in vascular biology.

Following vascular injury, HSP47 stabilizes the binding between platelets and collagen, enhancing platelet activation and aggregation while recruiting additional platelets to the injury site. In the wound healing process, HSP47 assists in the proper folding and secretion of procollagen, promoting healthy collagen deposition within the extracellular matrix (ECM).

Furthermore, HSP47 facilitates cell migration by binding to ECM components like fibronectin and laminin, which enhances granulation and vascularization. It also modulates interactions between collagens and immune cells, influencing the recruitment and activation of inflammatory cells essential for effective tissue regeneration.

Understanding the dual contributions of HSP47 in both thrombosis and wound healing can inform therapeutic strategies. Inhibiting HSP47 might exacerbate thrombus formation without hindering tissue repair, while enhancing HSP47 functionality could benefit both thrombotic events and the healing process.

HSP47 represents a promising target for drug development aimed at addressing bleeding disorders and improving recovery following vascular injuries. Ongoing studies of HSP47’s interactions within the ECM may reveal further therapeutic opportunities for managing disorders related to hemostasis and tissue repair [6,11,55].

#### 3.4.1. Unraveling HSP47’s Role in Collagen Dynamics: Future Research and Therapeutic Implications

Despite the insights gained from HSP47 knockout models, a complete understanding of the dependence of different collagen types on HSP47 remains elusive. More comprehensive research is needed to elucidate the mechanisms underlying HSP47 binding to various collagen types and the consequences of its absence in physiological contexts.

Future research directions include investigating the client-specific dependence of HSP47 on different collagen types, as some may require HSP47 more critically for proper folding and stability than others. Additionally, characterizing less understood collagen types, such as FACIT and MACIT subtypes, may identify new HSP47 clients and their physiological relevance. Employing advanced genetic and biochemical techniques, such as CRISPR/Cas9 and proteomic analyses, could provide deeper insights into the functional roles of HSP47 across various tissues and conditions [91].

Understanding HSP47’s interactions with collagen is essential for developing therapeutic strategies aimed at enhancing collagen stability in connective tissue diseases. Identifying small molecules that can promote HSP47 activity or strengthen its binding to specific collagen types could lead to new treatment approaches. Collectively, the binding properties of HSP47 are vital for maintaining tissue homeostasis and preventing disease, highlighting the need for further research to explore these complex interactions and their therapeutic potential in connective tissue disorders [1,31,91].

#### 3.4.2. Heat Shock Protein 47: A Dual Regulator of Platelet–Collagen Interactions and Immune Response in Plaque Thrombosis and VTE

Heat shock protein 47+ is a collagen-specific chaperone that plays a critical role in plaque thrombosis by enhancing platelet–collagen interactions and modulating immune cell activity. By maintaining collagen integrity, HSP47 ensures proper platelet adhesion and aggregation. Studies using knockout models and pharmacological inhibition have demonstrated that downregulation of HSP47 leads to impaired thrombus formation, longer bleeding times, and reduced platelet aggregation.

Additionally, decreased HSP47 expression can alter immune cell activities by modifying cytokine secretion and preventing immune cell influx, which delays inflammation. In the context of venous thromboembolism (VTE), chronic low-grade inflammation may lower HSP47 levels, impairing collagen remodeling and thrombus stability.

Alterations in HSP47 expression in patients with VTE correlate with risk factors such as immobility and obesity, emphasizing its potential as a therapeutic target for VTE management. Manipulating HSP47 levels may influence thrombus stability and aid in inflammation resolution. These findings suggest that HSP47 plays a dual role in hemostasis and inflammation, providing novel insights for pharmacological targeting in thrombotic disorders [6,91].

#### 3.4.3. Investigative Journey of Manuela Thienel

Manuela Thienel, a cardiologist, usually works in a hospital but ventured into a snowy Swedish forest in 2019 to study hibernating bears. She sought answers to why bears do not develop blood clots during hibernation, unlike immobile humans. After years of research involving various animals and human volunteers, her team identified a protein, HSP47, which decreases in bears during hibernation and may reduce clotting risk in humans. During hibernation, bears conserve energy by lowering their body temperature and slowing their metabolism. This likely reduces the need for HSP47, a protein that helps with collagen production, since tissue repair slows down. Hormonal changes, like lower insulin or growth factor levels, may also decrease HSP47 production. With less HSP47, blood clotting could decrease due to reduced collagen-related platelet activity, potentially protecting against clots during long periods of inactivity. Studying this could help find ways to lower clotting risks in humans, such as in deep vein thrombosis or stroke. This discovery suggests a potential new target for clot-preventing drugs, although further research and clinical trials are needed. Coagulation experts are intrigued by the unconventional approach taken by Thienel’s team in addressing a major medical challenge.

Dr. Manuela Thienel’s research explores the unique physiological adaptations of hibernating bears, particularly their ability to enter hibernation without developing blood clots. This investigation employs several methodologies to study protein expression during hibernation.

Proteomic analysis: mass spectrometry-based proteomic analysis of muscle tissue samples was conducted to identify differences in protein expression between active and hibernating bears.

Comparative studies: the research team compared the function and expression of HSP47 across various animal species and in human volunteers.

Functional analysis of blood coagulation: blood assays revealed that HSP47 levels correlate with coagulation efficiency, suggesting its role in preventing clot formation during hibernation.

Challenges in drug development: while HSP47 presents an exciting target for drugs aimed at preventing blood clotting, challenges remain, including ensuring specificity and safety of the drugs, addressing species variability, designing appropriate clinical trials, and navigating regulatory requirements for safety and efficacy testing.

Dr. Thienel’s research lays the foundation for developing novel strategies to manage clotting disorders, necessitating further investigation into HSP47 as a potential therapeutic target in clinical settings [2,92].

## 4. Discussion

This review underscores the emerging significance of heat shock protein 47 (HSP47) in thrombotic disorders, particularly deep vein thrombosis (DVT) and atherosclerosis. Traditionally recognized for its role as a collagen-specific chaperone in the endoplasmic reticulum (ER), HSP47 also facilitates platelet–collagen interactions critical for thrombus formation and vascular stability [1,5,6].

Recent studies demonstrate that inhibition or genetic deletion of HSP47 specifically reduces collagen-mediated platelet aggregation, calcium mobilization, and thrombus volume, without affecting thrombin-induced pathways [6,52,88]. This specificity suggests a therapeutic window for HSP47 modulation that could minimize thrombosis risk while preserving overall hemostasis.

Findings from hibernating mammals further support HSP47’s role in thrombo-protection. Bears, which experience prolonged immobility during hibernation, show decreased HSP47 expression and reduced thrombus formation, potentially due to lower collagen-dependent platelet activation and inflammation [2,6,93]. These natural adaptations provide a foundation for new antithrombotic strategies in immobile or critically ill patients.

In atherosclerosis, HSP47 influences collagen content and fibrous cap stability in plaques, thereby modulating rupture risk and cardiovascular outcomes [4,37]. Depending on disease context, both inhibition and enhancement of HSP47 activity may have therapeutic value—either by preventing excessive collagen accumulation in fibrosis or by reinforcing ECM stability in vulnerable plaques.

Despite promising evidence, challenges remain. Redundancy in platelet receptors (e.g., GPVI, integrin α2β1) and species-specific differences in HSP47 function necessitate careful therapeutic design [78,79]. Moreover, long-term systemic modulation of HSP47 could impact wound healing or connective tissue integrity.

Future studies should focus on clarifying collagen-type specificity, shear-dependent activity, and the interplay between HSP47 and immune signaling. The development of selective inhibitors or enhancers, combined with biomarker-based risk stratification, may unlock the full potential of HSP47-targeted therapies for thrombotic and connective tissue disorders.

To further contextualize the role of HSP47 in thrombotic disorders, Table 3 presents a comparative overview of HSP47 expression patterns and functional consequences across various physiological and pathological states. This includes data from both animal models such as hibernating bears and knockout mice and human conditions, including obesity, immobility, and atherosclerosis. Such comparisons underscore the translational relevance of HSP47, highlighting its potential as both a therapeutic target and a biomarker in diverse clinical settings.

## 5. Conclusions

In conclusion, HSP47, a key collagen-specific molecular chaperone, plays a vital role in collagen maturation, folding, and stabilization in humans. Its activity is crucial for maintaining the stability and assembly of collagen types essential to connective tissue health. While primarily functioning within the ER, HSP47’s influence extends beyond cellular protein synthesis, impacting adiposity regulation and cardiovascular health.

Research has highlighted its involvement in conditions such as DVT, atherosclerosis, and platelet-induced blood clotting. Particularly intriguing is the discovery of HSP47’s decreased expression in hibernating bears, potentially informing human therapeutic approaches for blood clot prevention in immobilized patients.

These studies suggest that modulating HSP47 could offer novel treatment pathways for connective tissue disorders and contribute to therapeutic innovations targeting clotting irregularities. Future research is essential to fully unearth HSP47’s mechanisms and therapeutic potential across different biological contexts.

Mirta Schattner at CONICET, Buenos Aires, described a breakthrough study on hibernating bear resistance to blood clotting, headed up by Ole Frobert, as “amazingly new”. It is one part of a growing trend toward seeking medical breakthroughs from non-model organisms, facilitated by next-generation sequencing and molecular tools. The research isolated heat shock protein 47 as responsible for preventing clotting, contrary to earlier assumptions regarding its role. Using techniques including whole transcriptome sequencing, proteomics, imaging (FRET, PET), single-cell RNA sequencing, and bioinformatics, scientists discovered that reduced HSP47 levels in hibernating bears inhibit clotting neutrophil extracellular traps. The mechanism appears translatable to human beings, as identical HSP47 reductions were observed in immobilized patients, zero-gravity simulations, and postpartum pigs. In hibernating bears, reduced HSP47 levels inhibit clotting by suppressing neutrophil extracellular trap (NET) formation, possibly by limiting inflammation and collagen signaling in their hypometabolic state, aided by unique physiological adaptations like altered blood viscosity. In immobilized patients with similar HSP47 reductions, clots still form due to the absence of these protective adaptations, allowing venous stasis, endothelial injury, and hypercoagulability to drive thrombosis, potentially via HSP47-independent NET pathways triggered by inflammation or damage [92,93].

These findings offer novel therapeutic potential for clotting diseases, such as venous thromboembolism, and perhaps for thrombosis-risk patients with cancer. HSP47’s role stabilizing collagen and regulating clotting makes it a potential drug target as well as a thrombotic risk biomarker. Challenges are off-target effects on platelets, regulation of clotting responses, and variability in protein expression. Future research will include validating HSP47 throughout hibernating animals, elucidating its mechanisms, and defining its uses for aging, chronic inflammation, and oncology, where it may regulate drug delivery and tumor microenvironments. Despite financial barriers, this nature-inspired strategy holds potential for advancing human health.

Therapeutic development: strategies might include pharmacological enhancers of HSP47, specific inhibitors in fibrotic conditions, gene therapy to increase HSP47 expression, and biologics targeting tissue repair. Combination therapies with existing anticoagulants or anti-inflammatory treatments could maximize efficacy by addressing multiple pathways involved in clotting and inflammation (Table 4).

In summary, HSP47 emerges as a significant target for disease-modifying therapies in thrombotic disorders and offers promising avenues for improving vascular health. Continued research on its mechanisms and therapeutic applications has the potential to enhance clinical outcomes for patients at risk of clotting abnormalities.

## Figures and Tables

**Table 1 cimb-47-00283-t001:** Summary of HSP47 Functions Across Biological Contexts and Disease Mechanisms.

Category	Key Points
HSP47 in Collagen Dynamics	- Acts as a chaperone for procollagen, ensuring proper folding and stability in the ER. - Prevents misfolding and aggregation of collagen molecules.
Role in Platelet Function	- Facilitates platelet adhesion and aggregation by stabilizing platelet-collagen interactions. - Key player in thrombus formation and stability.
Therapeutic Potential	- Targeting HSP47 offers potential for managing DVT, thrombosis, and connective tissue disorders. - Promising applications in wound healing and fibrosis.
Hibernation and Clot Resistance	- Decreased HSP47 levels in hibernating bears reduce clot formation. - Provides a model for developing antithrombotic drugs for immobilized patients.
Research Techniques	- Proteomics, RNA sequencing, and in vitro/in vivo models elucidate HSP47’s role. - Functional assays confirm its influence on platelet binding to collagen.
Adipose Tissue Function	- Influences collagen turnover and ECM integrity in adipose tissue. - Variability in HSP47 levels linked to obesity and fat distribution.
Atherosclerosis and Plaque Stability	- Modulates collagen stability in plaques. - Targeting HSP47 could enhance plaque stability and reduce cardiovascular risks.
Applications in Vascular Health	- Potential for novel therapies targeting HSP47 in clotting disorders, obesity, and inflammation. - Supports tissue repair and vascular remodeling.
Future Research Directions	- Explore HSP47’s role in ECM interactions, immune response, and inflammation. - Develop specific drugs targeting HSP47 pathways for disease treatment.

**Table 2 cimb-47-00283-t002:** Stages of Collagen Biosynthesis and HSP47’s Role in Intracellular Processing.

Stage	Location	Key Events	Key Components
1. Transcription	Nucleus	Collagen genes (e.g., COL1A1) are transcribed into mRNA.	Collagen genes, RNA polymerase
2. Translation	Rough ER ribosomes	mRNA is translated into alpha chains, which enter the ER lumen.	Ribosomes, mRNA, translocon
3. ER Processing	Endoplasmic Reticulum	Hydroxylation of proline and lysine (post-translational modification). Glycosylation of hydroxylysine. Three alpha chains form a triple helix (procollagen), stabilized by HSP47 to prevent misfolding.	Prolyl 4-hydroxylase, lysyl hydroxylase, vitamin C, HSP47
4. ER-to-Golgi Transport	ER to cis-Golgi	Folded procollagen is packaged into elongated COPII vesicles and transported along microtubules to the cis-Golgi; HSP47 remains bound.	COPII vesicles, TANGO1, microtubules, kinesin
5. Golgi Processing	Golgi Apparatus	HSP47 dissociates due to pH shift and recycles to ER via COPI vesicles. Minor glycosylation may occur. Procollagen is sorted into secretory vesicles.	COPI vesicles, secretory vesicles
6. Secretion	Plasma Membrane to ECM	Secretory vesicles fuse with the plasma membrane, releasing procollagen into the ECM.	SNARE proteins, calcium
7. ECM Maturation	Extracellular Matrix	Propeptides are cleaved by peptidases, forming tropocollagen. Tropocollagen self-assembles into fibrils. Lysyl oxidase cross-links fibrils into stable collagen fibers.	ADAMTS-2, BMP-1, lysyl oxidase

**Table 3 cimb-47-00283-t003:** Comparative Analysis of HSP47 Expression and Functional Outcomes in Physiological and Pathological States.

Species/Model	Physiological State	HSP47 Expression Profile	Functional Consequence	Relevance to Human Disease
Hibernating bears	Prolonged immobility	Downregulated	Decreased clot formation; lower platelet-collagen reactivity	Model for thrombo-protection during inactivity
Healthy humans	Normal activity	Homeostatic expression	Maintains collagen turnover and vascular integrity	Baseline for therapeutic modulation
Immobilized patients	Post-surgery, ICU, etc.	Variable; often dysregulated	Increased clotting risk due to inflammation and stasis	Target for clot-prevention therapies
Obese humans	Chronic inflammation	Upregulated in adipose tissue	Promotes fibrosis, collagen accumulation	Linked to metabolic syndrome and cardiovascular risk
Atherosclerotic plaques	Chronic vascular damage	Heterogeneous (↑ in fibrous cap, ↓ in necrotic core)	Affects plaque stability; promotes collagen cross-linking	Target for plaque stabilization or regression therapy
Mouse knockout models	HSP47-deficient	Absent	Impaired collagen synthesis, lethal phenotypes	Confirms essential role in connective tissue integrity

↑ high, ↓ low.

**Table 4 cimb-47-00283-t004:** Therapeutic Strategies Targeting HSP47 Across Clinical Conditions.

Biological Context	Therapeutic Approach	Potential Benefits	Challenges/Considerations	References
Collagen Stability (Connective Tissue Disorders)	Enhance HSP47 activity (small molecules, gene therapy)	Improved collagen folding and stability in disorders like osteogenesis imperfecta and Ehlers–Danlos syndrome	Risk of excessive collagen deposition leading to fibrosis; specificity to target collagen types	[4]
Fibrosis	Inhibit HSP47 (small-molecule inhibitors like SMIH)	Reduce excessive collagen accumulation in fibrotic conditions (e.g., pulmonary fibrosis, liver fibrosis)	Potential disruption of normal collagen synthesis; off-target effects on other tissues	[94]
Thrombosis (DVT, Atherosclerosis)	Inhibit HSP47 (selective inhibitors, antibodies)	Decrease platelet-collagen interactions, reducing thrombus formation and plaque rupture risk	Balancing hemostasis to avoid bleeding risks; specificity to collagen-mediated pathways	[6]
Atherosclerotic Plaque Stability	Modulate HSP47 (enhance or inhibit depending on context)	Enhance collagen content in fibrous caps for plaque stability; reduce inflammation	Need for precise control to avoid over- or under-stabilization; long-term safety unclear	
Wound Healing/Tissue Repair	Enhance HSP47 functionality	Promote proper collagen deposition and ECM remodeling for effective healing	Potential for excessive scarring or fibrosis if overexpressed	[95]
Venous Thromboembolism (VTE)	Modulate HSP47 levels (inhibition or enhancement)	Prevent thrombus formation in immobile patients; improve inflammation resolution	Variability in patient response; need to address underlying inflammation causes	[2]
Cardiovascular Risk Biomarker	Monitor HSP47 levels	Use as a biomarker for plaque stability or thrombotic risk stratification	Requires validation across populations; correlation with clinical outcomes needs study	[1]
Cancer (Tumor Microenvironment)	Target HSP47 for drug delivery or inhibition	Regulate collagen in tumor ECM; enhance drug delivery to tumors	Off-target effects on normal tissues; complexity of tumor-specific collagen dynamics	[96]
Bleeding Disorders	Enhance HSP47 activity	Strengthen platelet-collagen interactions to improve hemostasis	Risk of promoting unwanted thrombosis; individual variability in response	[6]
Combination Therapies	Combine HSP47 modulation with existing treatments (statins, anticoagulants)	Synergistic effects on collagen stability, inflammation, and clotting control	Drug interaction risks; need for clinical trials to optimize dosing and efficacy	[53]

## Data Availability

Data is contained within the article.

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
