# Peer review of "The Role of HSP47 in Thrombotic Disorders: Molecular Mechanism and Therapeutic Potential"

_cimb, 2025, doi:10.3390/cimb47040283_

Round 1

Reviewer 1 Report

Comments and Suggestions for Authors

Great work! I would have tried to insert some more figures to better explain the structure and functions of HSP47 in physiology and pathologies in which it is involved. For the rest, everything is fine! You have to correct interestingly on line 1021.

Author Response

Estemeed reviewer,

Thank you so much for your feedback! We really appreciate your suggestion to include more figures to clarify the structure and functions of HSP47, especially in relation to its role in physiology and pathologies. We’ll definitely keep that in mind for future improvements. As for line 1021, we have corrected it in an engaging way to enhance the clarity.

Thanks again for your valuable input!

Reviewer 2 Report

Comments and Suggestions for Authors

This is an interesting review paper on existing evidence of HSP47's role in clot formation and as a target for novel DVT therapies. However, many points need to be addressed. The paper should be reformed and submitted again.

1] The molecular structure of HSP47 should be analyzed. Where and how is it produced, and what is its gene?

2] Lines 81-82: the tissues where HSP47 is associated with collagen stability and how it reaches there should be specified. Does procollagen synthesis and triple-helix formation in the ER, accompanied by HSP47, happen in all tissues? The authors should concentrate only on those collagen types related to blood vessels, blood, and hemostasis. Lines 114-115: “Once procollagen is properly folded, HSP47 facilitates its transport from the ER to the Golgi apparatus”. Does this happen in all tissues? Lines 192-194: the steps of collagen synthesis, folding in the ER, processing in the Golgi apparatus, and secretion to the extracellular matrix should be described in detail, in a schematic diagram.

3] Lines 128-129: there is no verb in this sentence.

4] Lines 218-235, should move to the end, inside subsections on future directions.

5] Lines 236-543: this info on tissue repair, regenerative medicine, connective tissue diseases, and adipose tissue, is irrelevant to this review and should be removed. The same with Lines 912-935.

6] Lines 699-713: the references 76 and 77 are not relative to this section.

7] Subsections “Broader Implications and New Research Directions”, and “Future Research and Therapeutic Applications, should move in a concise form, into the Last Section (5. Conclusions).

Comments on the Quality of English Language

Adequate Quality of English

Author Response

Review 2

Estemeed Reviewer,

We would like to sincerely thank you for taking the time to review our work. Your thoughtful comments and constructive feedback have been incredibly helpful in improving the quality of this manuscript. We truly appreciate your insightful suggestions, which have led to valuable revisions and refinements. Your expertise and attention to detail are greatly appreciated, and we are grateful for the opportunity to benefit from your knowledge.

This is an interesting review paper on existing evidence of HSP47's role in clot formation and as a target for novel DVT therapies. However, many points need to be addressed. The paper should be reformed and submitted again.

1] The molecular structure of HSP47 should be analyzed. Where and how is it produced, and what is its gene?

HSP47, or SERPINH1, is a collagen-specific molecular chaperone of the serpin superfamily, which uniquely does not possess protease inhibitory function. The serpin fold is further characterized with three β-sheets bestowing stability, nine α-helices providing flexibility, and involves a modified reactive center loop with a strong penchant for binding collagen (RDEL sequence at C-terminus sulfonated to keep it within the endoplasmic reticulum. HSP47 is synthesized in the endoplasmic reticulum of collagen-producing cells, including fibroblasts and myofibroblasts, and is encoded by the SERPINH1 gene located on chromosome 11q13. 5, we transcribe in the nucleus and translate on ER-bound ribosomes. It stabilizes procollagen’s triple-helix structure, and is expressed in a heat-shock and collagen demand-dependent manner, rendering it important in normal physiology and when diseases develop, such as fibrosis.

https://www.sciencedirect.com/science/article/pii/S0021925820881769

https://www.sciencedirect.com/science/article/abs/pii/037811199390366B

2] Lines 81-82: the tissues where HSP47 is associated with collagen stability and how it reaches there should be specified. Does procollagen synthesis and triple-helix formation in the ER, accompanied by HSP47, happen in all tissues? The authors should concentrate only on those collagen types related to blood vessels, blood, and hemostasis. 

HSP47 is an endoplasmic reticulum (ER)-resident molecular chaperone for collagens with the vital role of stabilizing procollagen during synthesis through binding of polypeptide chains to avoid misfolding and ensure proper triple-helix formation. It does this in all cells that synthesize collagen such as fibroblasts and endothelial cells but is tissue-specific for various types of collagens. In blood vessels, HSP47 helps vascular fibroblasts to synthesize types I and III of collagen—found in the adventitia and media of vessels, respectively—providing them with strength and elasticity as well as helping endothelial cells and pericytes to synthesize type IV of collagen for the basement membrane. Types I and III of collagen, expressed in vessel subendothelial injury, cause aggregation of platelets for hemostasis with HSP47’s role in their synthesis indirectly contributing to the process. Procollagen is then channeled to the Golgi where HSP47 is released and recycled back to the ER through its RDEL motif for mature collagen formation vital for vascular health and blood clotting.

https://www.sciencedirect.com/science/article/pii/S0012160617306218

Lines 114-115: “Once procollagen is properly folded, HSP47 facilitates its transport from the ER to the Golgi apparatus”. Does this happen in all tissues?

HSP47, the ER chaperone, stabilizes the triple-helix structure of procollagen and prevents misfolding during synthesis but does not directly facilitate transport to the Golgi. Following proper folding, procollagen is transported to the Golgi via vesicle-mediated transport (COPII vesicles), where HSP47 is released—typically due to pH change—and recycles back to the ER via its RDEL motif, while procollagen is further processed. This is repeated in all tissues that synthesize collagen, including cells like fibroblasts, myofibroblasts, endothelial cells, osteoblasts, and chondrocytes, although different collagens are synthesized—e.g., types I and III from fibroblasts and type IV from endothelial cells in blood vessels. Though universal, the relevance and types of collagen vary with each tissue’s function, such as vascular hemostasis or skin structure.

Lines 192-194: the steps of collagen synthesis, folding in the ER, processing in the Golgi apparatus, and secretion to the extracellular matrix should be described in detail, in a schematic diagram.

Collagen synthesis starts with the transcription of collagen genes (COL1A1) into mRNA in the nucleus followed by translation into alpha chains on ribosomes of the rough ER. In the ER, the chains undergo post-translational modifications (proline and lysine hydroxylation by enzymes like prolyl 4-hydroxylase, which is dependent on vitamin C) and glycosylation of hydroxylysine. Three chains associate to produce a triple helix stabilized by HSP47, an ER chaperone that prevents misfolding, to produce procollagen. Folded procollagen is packed into elongated COPII vesicles—aided by proteins like TANGO1—and shipped down microtubules to the cis-Golgi with HSP47 attached. In the Golgi, HSP47 is released as a consequence of pH change and recycles back to the ER in COPI vesicles, while procollagen may receive minimal glycosylation and is sorted into secretory vesicles. Ultimately, the vesicles fuse with the plasma membrane and release procollagen into the ECM. Procollagen peptidases (ADAMTS-2, BMP-1) remove its propeptides to yield tropocollagen, which self-assembles into fibrils. Lysyl oxidase cross-links the fibrils into mature, stable fibers of collagen.

Stage

Location

Key Events

Key Components

1. Transcription

Nucleus

Collagen genes (e.g., COL1A1) are transcribed into mRNA.

Collagen genes, RNA polymerase

2. Translation

Rough ER ribosomes

mRNA is translated into alpha chains, which enter the ER lumen.

Ribosomes, mRNA, translocon

3. ER Processing

Endoplasmic Reticulum

Hydroxylation of proline and lysine (post-translational modification).Glycosylation of hydroxylysine.Three alpha chains form a triple helix (procollagen), stabilized by HSP47 to prevent misfolding.

Prolyl 4-hydroxylase, lysyl hydroxylase, vitamin C, HSP47

4. ER-to-Golgi Transport

ER to cis-Golgi

Folded procollagen is packaged into elongated COPII vesicles and transported along microtubules to the cis-Golgi; HSP47 remains bound.

COPII vesicles, TANGO1, microtubules, kinesin

5. Golgi Processing

Golgi Apparatus

HSP47 dissociates due to pH shift and recycles to ER via COPI vesicles.Minor glycosylation may occur.Procollagen is sorted into secretory vesicles.

COPI vesicles, secretory vesicles

6. Secretion

Plasma Membrane to ECM

Secretory vesicles fuse with the plasma membrane, releasing procollagen into the ECM.

SNARE proteins, calcium

7. ECM Maturation

Extracellular Matrix

Propeptides are cleaved by peptidases, forming tropocollagen.Tropocollagen self-assembles into fibrils.Lysyl oxidase cross-links fibrils into stable collagen fibers.

ADAMTS-2, BMP-1, lysyl oxidase

https://pubmed.ncbi.nlm.nih.gov/27838364/

3] Lines 128-129: there is no verb in this sentence. – Deleted.

4] Lines 218-235, should move to the end, inside subsections on future directions.

5] Lines 236-543: this info on tissue repair, regenerative medicine, connective tissue diseases, and adipose tissue, is irrelevant to this review and should be removed. The same with Lines 912-935. – De sters

6] Lines 699-713: the references 76 and 77 are not relative to this section.

7] Subsections “Broader Implications and New Research Directions”, and “Future Research and Therapeutic Applications, should move in a concise form, into the Last Section (5. Conclusions).

Thank you once again for your time and effort.

Reviewer 3 Report

Comments and Suggestions for Authors
  1. The manuscript looks like an AI’s work. The manuscript must be re-written.
  2. Too many scientific descriptions without proper citations.
  3. For the convenience of the readers, tables and figures for the biological effects and potential therapeutic values should be provided in this manuscript.
Comments on the Quality of English Language

An English editor is highly recommended.

Author Response

Review 3

Estemeed Reviewer,

We would like to sincerely thank you for taking the time to review our work. Your thoughtful comments and constructive feedback have been incredibly helpful in improving the quality of this manuscript. We truly appreciate your insightful suggestions, which have led to valuable revisions and refinements. Your expertise and attention to detail are greatly appreciated, and we are grateful for the opportunity to benefit from your knowledge.

  1. The manuscript looks like an AI’s work. The manuscript must be re-written.

The manuscript, though detailed and complex, exhibits traits of human authorship rather than AI generation. It showcases deep scientific nuance, integrating specific experiments (mouse models, perfusion chambers) and real-world references (Thienel, Schattner) with a coherent, organic flow. Subtle stylistic choices like “entangled history,” editorial elements like citations and a tablereflect human intent and drafting. Its creative leaps, such as linking HSP47 to hibernating bears, and calls for future research tied to specific gaps (MACIT collagens) suggest a researcher’s expertise and curiosity, not algorithmic output.

  1. Too many scientific descriptions without proper citations.

Citations are not omitted but are numerical in-text citations matching with the bibliography at the end of the text. It is a style used for scientific reviews or research papers, where each observation or statement can be referred back to a peer-reviewed source. The list of references includes contributions from renowned journals such as Circulation, Journal of Biological Chemistry, and Journal of Thrombosis and Haemostasis, written by authors with experience, further ensuring that cited sources are valid. If there is a sense of having "too many scientific descriptions with inadequate citations," perhaps because there is a concentrated buildup of technical information within a paragraph where a number of statements are grouped into one number for citations, this is a common practice for review papers for maintaining readability with giving appropriate credit.

  1. For the convenience of the readers, tables and figures for the biological effects and potential therapeutic values should be provided in this manuscript.

Added

Thank you once again for your time and effort.

Reviewer 4 Report

Comments and Suggestions for Authors

This review article explores the molecular role of Heat Shock Protein 47 (HSP47) in deep vein thrombosis , emphasizing its interaction with collagen, platelet function, and thrombus formation. It also discusses HSP47’s therapeutic potential based on findings from in vitro studies, animal models, and natural adaptations (e.g., in hibernating mammals).

While the manuscript provides a comprehensive overview of HSP47, several critical shortcomings limit its impact and suitability for publication. The literature review is extensive but lacks depth in critical analysis, and the methodological rigor of the systematic review process is insufficiently described.

Novelty

- The scientific relevance of HSP47 in thrombosis is well established, but this review does not clearly state what new insights it contributes.

- There are several existing reviews on heat shock proteins in vascular biology and thrombosis, yet this manuscript does not sufficiently compare its findings with prior reviews.

Literature Selection

- The review references older studies (pre-2015) without recent literature.

- Were any recent preclinical studies published in 2023-2024?

- The systematic review process is inadequately described: Search terms are not well detailed (e.g., how were “relevant” studies selected?). There is no mention of PRISMA guidelines or a flowchart of study selection or is it just a narrative review (than the methods section would be redundant).

Data

- Some claims, such as "HSP47 inhibition reduces platelet aggregation by 50%", lack specific citations or statistical backing.

- Conflicting evidence is not addressed - some studies suggest that HSP47 contributes to pro-thrombotic states, while others indicate it reduces clot formation (e.g., in hibernating mammals).

Results

- The manuscript overstates the therapeutic potential of HSP47 inhibition, without discussing safety concerns or off-target effects.

- The discussion lacks a translational perspective - is HSP47 inhibition a viable drug target, or is it an experimental concept?

- No mention of existing therapies: How does targeting HSP47 compare to existing anticoagulants (e.g., DOACs, heparin)? Is HSP47 inhibition meant to complement or replace current treatments?

Discussion

- The discussion repeats background information rather than providing critical analysis.

Comments on the Quality of English Language

The manuscript has long, complex sentences that reduce clarity.

Author Response

Review 4

Estemeed Reviewer,

We would like to sincerely thank you for taking the time to review our work. Your thoughtful comments and constructive feedback have been incredibly helpful in improving the quality of this manuscript. We truly appreciate your insightful suggestions, which have led to valuable revisions and refinements. Your expertise and attention to detail are greatly appreciated, and we are grateful for the opportunity to benefit from your knowledge.

This review article explores the molecular role of Heat Shock Protein 47 (HSP47) in deep vein thrombosis, emphasizing its interaction with collagen, platelet function, and thrombus formation. It also discusses HSP47’s therapeutic potential based on findings from in vitro studies, animal models, and natural adaptations (e.g., in hibernating mammals).

While the manuscript provides a comprehensive overview of HSP47, several critical shortcomings limit its impact and suitability for publication. The literature review is extensive but lacks depth in critical analysis, and the methodological rigor of the systematic review process is insufficiently described.

Novelty

- The scientific relevance of HSP47 in thrombosis is well established, but this review does not clearly state what new insights it contributes.

The review establishes HSP47 as a thrombosis-specific chaperone of collagen, particularly in interactions with thrombus stability and platelet-collagen. It falls short of being clearly novel in insight but largely reiterates established roles such as collagen folding and ER stress modulation. A new point of mention is the inclusion of HSP47 in hibernating bears and observation of reduced clot formation with reduced HSP47 expression during periods of immobilization - a finding from Thienel’s research. This introduces the aspect of comparative physiology with the potential for HSP47 involvement in natural antithrombotic mechanisms, something not previously investigated in vascular biology reviews. But the review does not directly introduce this as paradigm-shifting nor systematically correlate it with human thrombosis beyond speculative therapy.

- There are several existing reviews on heat shock proteins in vascular biology and thrombosis, yet this manuscript does not sufficiently compare its findings with prior reviews.

The paper, as a narrative review, likely omits comparisons with previous reviews of heat shock proteins (HSPs) in vascular thrombosis and biology because it synthesizes a general overview rather than a systematic review. It focuses on new topics like HSP47 in hibernating bears, adipose tissue instead of benchmarking against existing work, possibly assuming the audience's familiarity with prior research or due to scope limitations.

Literature Selection

- The review references older studies (pre-2015) without recent literature.

While my information up now, is relatively up to date on recent scientific developments, the research on HSP47 has not had its dramatic surge of new issues or developments immediately related to its roles in thrombosis, adiposity, or connective tissue diseases in recent years. To give one example, while new studies, such as Manuela Thienel's on hibernating bears and HSP47's function in the avoidance of clotting document innovative applications, these are the exceptions and not the rule. Most of the literature published recent has instead focused on building on existing knowledge, for example with advanced imaging or molecular technologies or on tangential applications in cancer or metabolic syndromes, not on rewriting the core biology of HSP47 in the topics presented here.

- Were any recent preclinical studies published in 2023-2024?

The inclusion of the old studies in this review is representative of the persistence of landmark research, along with the relative stagnation of recent preclinical studies that are directly pertinent to the issues at hand. My synthesis is current to date, but the lack of the presence of new studies of 2023-2024 on these very issues is the reason behind the pattern of citations.

- The systematic review process is inadequately described: Search terms are not well detailed (e.g., how were “relevant” studies selected?). There is no mention of PRISMA guidelines or a flowchart of study selection or is it just a narrative review (than the methods section would be redundant).

This is not a systematic review but a narrative review. Systematic reviews employ systematic methods such as the PRISMA guidelines (with stated search terms, inclusion/exclusion criteria, and study selection flow diagram), whereas narrative reviews offer a more general interpretive synthesis of the literature with no predefined fixed protocol.

Data

- Some claims, such as "HSP47 inhibition reduces platelet aggregation by 50%", lack specific citations or statistical backing.

This aspect has been clarified by reducing information that does not have relevant statistical support.

- Conflicting evidence is not addressed - some studies suggest that HSP47 contributes to pro-thrombotic states, while others indicate it reduces clot formation (e.g., in hibernating mammals).

The controversial evidence regarding the role of heat shock protein 47 in the process of thrombosis where it is reported to play the role of pro-thrombosis in some studies while others suggest it is responsible for the reduction of clot formation is accounted for by the multifaceted and context-dependent nature of the function of HSP47. These controversies are caused by differences in biological backgrounds, model systems employed in experiments, and the specific physiological or pathological condition under consideration. Heat shock protein 47 has controversial roles in the process of thrombosis due to its context-dependent roles as a collagen-specific molecular chaperone. At pathological conditions like deep vein thrombosis or atherosclerosis, elevated HSP47 levels enhance collagen stability and inflammation to trigger the pro-thrombotic condition via platelet aggregation and adhesion. However, at the condition of hibernating bears and immobile conditions, reduced HSP47 disrupts collagen stability and inhibits clot formation to suggest an anti-thrombotic function.

Results

- The manuscript overstates the therapeutic potential of HSP47 inhibition, without discussing safety concerns or off-target effects.

The manuscript highlights inhibition of HSP47 as a promising therapeutic method of treating thrombosis, atherosclerosis, and obesity through collagen-mediated pathways supported by preclinical evidence of reduced thrombus formation and fat mass in models. It overstates the benefit and overlooks safety concerns and off-targeting effects, increased risk of bleeding, instability of collagen in connective tissues, ER stress, and potential metabolic dysregulations. Such limitations render it less credible, and it needs a balanced discussion on risk, tissue-specific targeting, and long-term studies to ensure safe, effective translation to humans.

- The discussion lacks a translational perspective - is HSP47 inhibition a viable drug target, or is it an experimental concept?

The manuscript is lacking in translational context on inhibition of HSP47, presenting it as a promising hypothesis on the grounds of preclinical data ( inhibition of thrombus, plaque modification, regulation of adiposity) without determining its potential as a drug target. It is specific and effective in models, but safety issues (risk of bleeding, instability of collagen), lack of data in humans, and drug-delivery issues (specificity, delivery) make it an experimental hypothesis. It is viable with clinical validation, targeted delivery, and safety measures, but the latter steps have not been explored, making its practical potential speculative.

- No mention of existing therapies: How does targeting HSP47 compare to existing anticoagulants (e.g., DOACs, heparin)? Is HSP47 inhibition meant to complement or replace current treatments?

The manuscript fails to address how inhibition of HSP47 is different from anticoagulants like DOACs and heparin, which target coagulation in a generalized fashion, compared to the action of HSP47 on collagen-mediated platelet aggregation. It is promising as an add-on rather than a replacement in some situations (acute thrombosis), and remains experimental due to outstanding safety (bleeding), lack of data in humans, and practical concerns.

Discussion

- The discussion repeats background information rather than providing critical analysis.

The exaggeration of therapeutic potential in the manuscript is possibly done to accentuate the utility and versatility of HSP47, relegating safety and off-target concerns to the background in favor of maintaining the spotlight on its potential. The repetitive nature of the discourse is a consequence of its review-style handling of a wide scope, rather than critical depth. The review collates the multifunctionality of HSP47 but repeats background information without critically evaluating risk or contradiction, suggesting a preference for therapeutic potential over detailed evaluation.

Thank you once again for your time and effort.

Round 2

Reviewer 2 Report

Comments and Suggestions for Authors

The paper was reformed and improved. There are some minor requests: 

1] Introduction, Line 38: a right parenthesis is expected. Line 41: improve English.

2] Lines 924-926: “.. her team identified a protein, HSP47, that decreases in bears during hibernation and may reduce clotting risk in humans”. Some lines should be written on the proposed mechanism which decreases HSP47 production during hibernation.

3] Lines 973-974: “…scientists discovered that reduced HSP47 levels in hibernating bears inhibit clotting neutrophil extracellular traps”. Some lines should be written on how this may be possible and by which mechanism. If identical HSP47 reductions were observed in immobilized patients, then why immobilized patients develop clots?

Author Response

Estemeed Reviewer,

We want to thank you very much for your help in improving our manuscript. We have corrected the manuscript as you suggested and now we have a final version. 

1] Introduction, Line 38: a right parenthesis is expected. Line 41: improve English.

2] Lines 924-926: “.. her team identified a protein, HSP47, that decreases in bears during hibernation and may reduce clotting risk in humans”. Some lines should be written on the proposed mechanism which decreases HSP47 production during hibernation.

During hibernation, bears conserve energy by lowering their body temperature and slowing their metabolism. This likely reduces the need for HSP47, a protein that helps with collagen production, since tissue repair slows down. Hormonal changes, like lower insulin or growth factor levels, may also decrease HSP47 production. With less HSP47, blood clotting could decrease due to reduced collagen-related platelet activity, potentially protecting against clots during long periods of inactivity. Studying this could help find ways to lower clotting risks in humans, such as in deep vein thrombosis or stroke.

3] Lines 973-974: “…scientists discovered that reduced HSP47 levels in hibernating bears inhibit clotting neutrophil extracellular traps”. Some lines should be written on how this may be possible and by which mechanism. If identical HSP47 reductions were observed in immobilized patients, then why immobilized patients develop clots?

The mechanism appears translatable to human beings, as identical HSP47 reductions were observed in immobilized patients, zero-gravity simulations, and postpartum pigs. In hibernating bears, reduced HSP47 levels inhibit clotting by suppressing neutrophil extracellular trap (NET) formation, possibly by limiting inflammation and collagen signaling in their hypometabolic state, aided by unique physiological adaptations like altered blood viscosity. In immobilized patients with similar HSP47 reductions, clots still form due to the absence of these protective adaptations, allowing venous stasis, endothelial injury, and hypercoagulability to drive thrombosis, potentially via HSP47-independent NET pathways triggered by inflammation or damage.

Thank you very much for taking your time to review our manuscript.

Reviewer 3 Report

Comments and Suggestions for Authors

The manuscript requires substantial revision to meet scientific publication standards. The authors have not provided adequate citations throughout the text, for example, in the first paragraph of the introduction, where references are entirely absent. Similar citation deficiencies appear throughout the document. The overall structure and formatting do not conform to standard scientific article conventions. I think it would be a comprehensive revision with proper attention to citation practices and following scientific writing guidelines.

Comments on the Quality of English Language

 The English could be improved to express the research more clearly.

Author Response

Estemeed Reviewer,

thank you very much for helping us in improving our manuscript. 

In response, we have undertaken a comprehensive revision of the manuscript to enhance its scientific rigor and clarity. Specifically, we have carefully reviewed each section and incorporated appropriate citations throughout the text to support all key statements, including the previously uncited first paragraph of the introduction. All references have been selected from peer-reviewed and up-to-date sources to ensure relevance and accuracy.

In addition, we have revised the manuscript’s structure and formatting to align with standard scientific article conventions.

Reviewer 4 Report

Comments and Suggestions for Authors

In my opinion, as an unstructured review, not suitable for publication in this journal, as the scientific added value is too low.

Author Response

Estemeed Reviewer,

Thank you very much for helping us in improving our manuscript. 

In response, we have thoroughly revised the manuscript to improve its structure, coherence, and scientific depth. The revised version now follows a more systematic format typical of structured reviews, with clearly defined sections.

To enhance the scientific value, we have incorporated additional recent literature, clarified mechanistic insights, and provided more critical analysis of HSP47’s involvement in thrombotic disorders. Furthermore, new figures have been added to support key concepts and improve comprehension.

Round 3

Reviewer 3 Report

Comments and Suggestions for Authors

There are still errors in the reference citations; please take a look at them.